# Assessment of Mechanical Properties of Corroded Prestressing Strands

**Chi-Ho Jeon, Cuong Duy Nguyen and Chang-Su Shim \***

A Department of Civil and Environmental Engineering, Chung-Ang University, Seoul 06974, Korea; chihobeer@cau.ac.kr (C.-H.J.); nguyencuong2892@cau.ac.kr (C.D.N.)

**\*** Correspondence: csshim@cau.ac.kr; Tel.: +82-2-820-5895

**Abstract:** The corrosion of prestressing steel in prestressed concrete bridges is a critical issue for bridge maintenance. To assess structures with corroded strands, it is necessary to define the mechanical properties of the strands and their influence on the structural behavior. In this study, corroded strands were taken from external tendons in existing post-tensioned concrete bridges and tested to determine the effects of corrosion on their tensile properties. Empirical equations for the tensile strength and ductility of the corroded strands were proposed using test results. The most corroded wire governs the mechanical properties of the strand. Experiments on prestressed concrete beams with a single corroded strand were conducted to investigate their structural behavior. A reduction in the flexural strength and maximum deformation was observed in these experiments. According to the section loss of a wire in a strand and its location in a beam, the flexural capacity can be evaluated using the proposed equation. The reduced ultimate strain of the corroded strand can be the governing factor of the flexural strength.

**Keywords:** corrosion; prestressed concrete bridge; prestressing steel; section loss; strength; ductility

---

## 1. Introduction

The tendon is the most important structural element in a prestressed concrete (PSC) girder, and its damage can significantly influence the girder's global behavior. The most critical type of damage in tendons is corrosion, which is difficult to detect through inspection. Corrosion-induced failure is not considered in the design of PSC structures and may cause the sudden collapse of a bridge. Once corrosion is found in a tendon, it is a difficult task to quantify the level of corrosion and its location through the entire length of the tendon. Some of the multiple strands in a tendon are invisible and cannot be observed. Therefore, a method to evaluate the level of corrosion as well as the corresponding mechanical properties of such corroded strands is needed.

In the bridges mentioned in a recent American report [1,2], such as the Niles Channel Bridge, Mid-Bay Bridge, Bob Graham Sunshine Skyway Bridge, and Varina-Enon Bridge, external tendon corrosion occurred within the last 20 years of a relatively short service period. Its causes were mainly infiltration of chloride-contaminated water, insufficient grout filling, and uncertain anchorage closure. When these factors occur together, severe corrosion may result. Uncertain anchorage closure could allow the entrance of water, air, and chloride, and imperfect grout filling may permit the exposure of internal strands to pollutants, causing corrosion, as shown in Figure 1. According to the case study of Carsana et al. [3], one bridge with tendon fractures caused by corrosion had enough grout filling but, owing to the segregation of the grout, severe tendon corrosion occurred within 2 years. Chloride was not detected; however, a high sulfate ion content was found on the surface of the grout, and it was assumed to be the cause of the weakening of the grout segregation and strength.

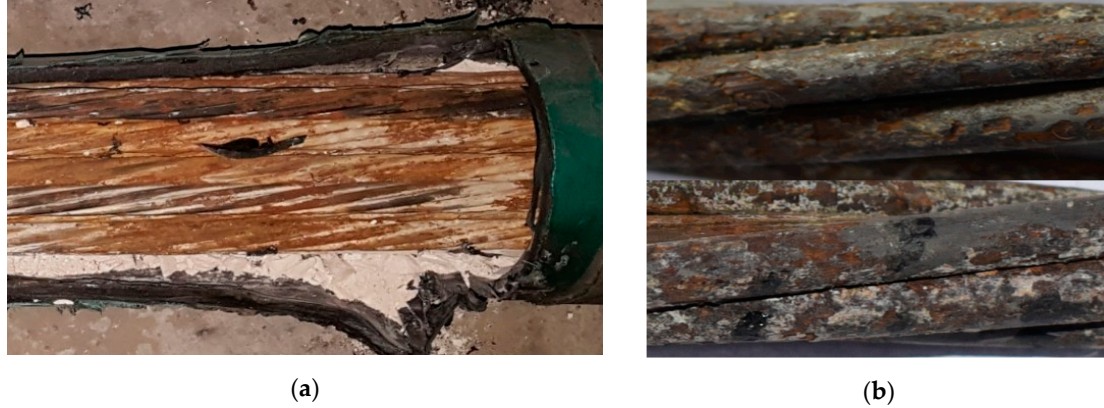

(**a**) (**b**)

**Figure 1.** Observed corroded strands in existing bridges: (**a**) corroded tendon; (**b**) corroded strand.

Inspection of corroded strands is mostly been done for external tendons, as it is easier to observe and replace these. Corrosion occurring in inner tendons is difficult to repair even when detected. Therefore, most reports mentioned above and the inspection methods suggested therein focused on the corrosion of external tendons. Several studies [4–6] have been conducted to evaluate the section loss caused by the corrosion of a strand and the subsequent change in mechanical properties. In all of these research studies, the level of corrosion was evaluated using one of two methods.

The first method is to use the loss of mass to evaluate the section loss and apply it to the mechanical property evaluation. This approach uses electrochemical methods to induce the corrosion of a strand. The mass loss based on Faraday's law for the evaluation of the chemical reaction of the metal is used as the specimen's damage index in relation to the corrosion. This approach was introduced in ASTM G1-03 from American Society for Testing and Materials (ASTM) [7]. These studies obtained similar results for the decreases in strength and ductility caused by the progression of corrosion. However, there is no viable method to evaluate the corrosion on the strands of an existing PSC bridge. Evaluating the level of corrosion using the mass is not suitable.

The second approach is to use a pit-depth gauge to measure the depth of the corrosion pit. The section loss is evaluated using the methods to evaluate pitting given in ASTM G49-94 [8] regarding corrosion. As this approach measures the depth of the corrosion pit, an idealization of the section loss model is needed to quantify the section loss. Many researchers [9–14] have suggested section loss models such as hemispherical and planar pit configurations. Previous research [15] suggested three types of corrosion and their associated section losses through the observation of corroded strands from PSC bridges. With the help of tests and analyses, the mechanical properties of the corroded strands were determined.

In this study, inspections and tensile experiments were conducted on 100 corroded strand specimens obtained from bridges in service, which were also investigated in previous studies [15,16]. The difference from the previous research [15] is that the mechanical properties of the corroded strands were analytically obtained in the previous study; however, this study performed statistical analysis based on the experiment for practical use, and applied the results to predict flexural strength. Based on the experimental results, a reduction in the ultimate strength and fracture strain is suggested in the form of an empirical formula. Flexural tests were also conducted on PSC beams with corroded strands. The behavior of the corroded beams was evaluated, and the suggested equations were verified.

## 2. Investigation of Corrosion in External Tendons

### 2.1. Bridge Description

Two post-tensioned segmental box girder bridges, referred to as "A" and "B", located in an urban arterial road and that were known to have corrosion in their external tendons were examined. Bridge A used 19 seven-wire strands with a nominal diameter of 15.2 mm, while Bridge B used 27 seven-wire

strands with a nominal diameter of 12.7 mm. A fracture in one external tendon of Bridge A was found 18 years after the bridge construction was completed. The external tendons were covered by a duct filled with grout. To decide on the repair or replacement of the corroded tendons, the ducts were uncovered, and a detailed inspection was conducted. Severe local section losses were found on the corroded strands. The investigation concluded that the causes were poorly treated air vents and joints in the duct resulting in an infiltration of chloride, presumed to be from de-icing chemicals, and rainwater. In addition, voids not filled by the grout were found inside the duct, and most of the corrosion was located in those sections. In the tendons that were selected for replacement, the corroded strands were cut and moved to a lab to evaluate the corrosion. Tensile tests were conducted on 100 corroded strands.

### 2.2. Corrosion Inspection

The corrosion on the strands along the length had very irregular pits, which made it difficult to determine the worst part of the corroded wire by visual inspection. To quantify the corrosion of the specimen, the ASTM G46-94 [8] corrosion measurement method was used, as introduced in a previous study [15]. This method uses a pit-depth gauge (Mitutoyo Corporation, Kanagawa, Japan) to measure the depth of the pit, as shown in Figure 2; however, this method is unable to evaluate the section loss. Therefore, as in the previous research, a section-loss evaluation method using the pit depth was suggested.

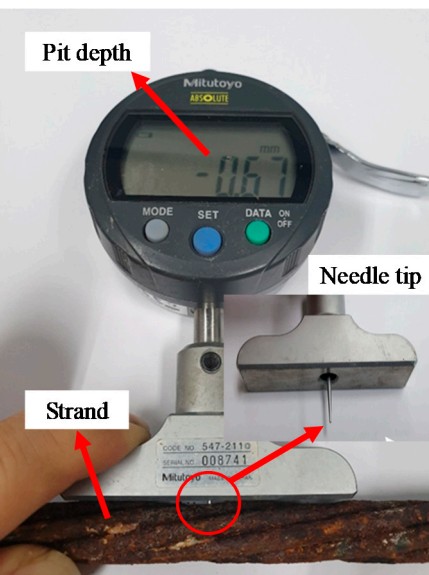

**Figure 2.** Measuring pit depth of corroded wire.

Figure 3 illustrates five cases of corroded strands located near the anchorage and their cut sections. The samples had different levels of corrosion, and the corrosion pits were easily observed when the corrosion is severe. The five samples were divided into corroded and non-corroded parts. Because a high number of strands exist in a tendon bundle, one portion is exposed to the air and another portion is in contact with neighboring strands. This results in the corrosion being concentrated on several strands. These samples were not the specimens used in the tensile test, because the cut section pictures could not be taken before the tensile test.

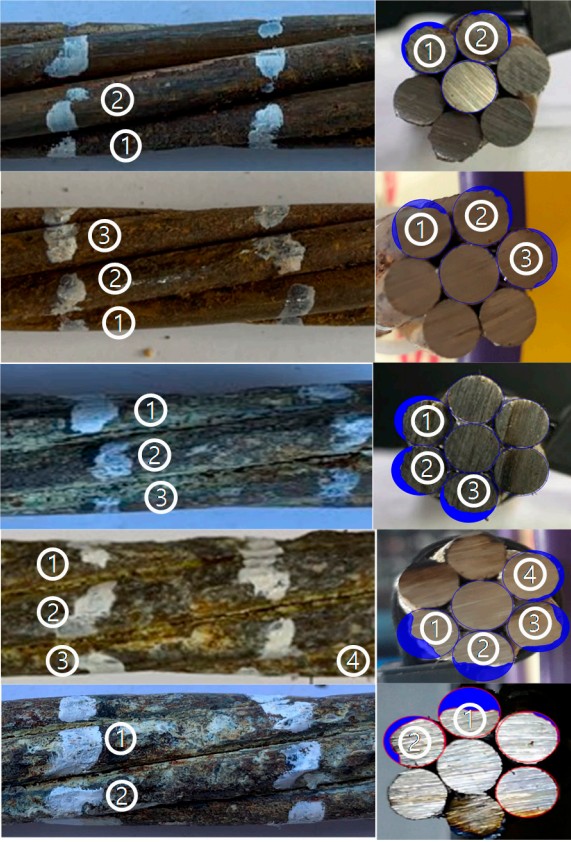

**Figure 3.** Corrosion of a strand near the anchor in Bridge B.

Figure 4 shows an idealized model for the evaluation of each wire's actual section loss pattern and the section loss suggested by previous research [15]. Equations (1)–(3) were used for the evaluation of the section loss for each pattern, using only the measured pit depth of the corroded strands. This approach does result in errors between the actual and measured corrosion values, but it is the most feasible method for an onsite inspector to evaluate the corrosion, and therefore it was used in this research. The remaining corrosion residue on the extracted specimen was removed by a corrosion removal solution. Next, along the strand's length, the depth of the corrosion was measured every 20 mm using a pit-depth gauge, as shown in Figure 5. Excluding the core wire, which could not be evaluated for section loss, the degree of corrosion of each strand was evaluated as the summation of each section loss of the remaining six wires, as given in Equations (1)–(3). The highlighted part in Figure 5 is the section that was determined to be the most corroded (the highest section loss) by comparing the summations of the section losses.

$$A_{sl,1} = 2r^2 \left[ \arccos\left(1 - \frac{d_p}{2r}\right) - \frac{1}{2}\sin\left\{2\arccos\left(1 - \frac{d_p}{2r}\right)\right\} \right] \tag{1}$$

$$A_{sl,2} = r^2 \left[ 2\left\{\arccos\left(-\frac{d_p}{2r}\right)\right\} - \pi - \sin\left\{2\arccos\left(-\frac{d_p}{2r}\right)\right\} \right] \tag{2}$$

$$A_{sl,3} = r^2 \left[ \arccos\left(1 - \frac{d_p}{r}\right) - \frac{1}{2}\sin\left\{2\arccos\left(1 - \frac{d_p}{r}\right)\right\} \right] \tag{3}$$

where $A_{sl,1-3}$ are the losses of sectional areas according to the type of pit configuration, $r$ is the radius of the wire, and $d_p$ is the pit depth (from 0 to $2r$) measured by a depth gauge at the deepest location.

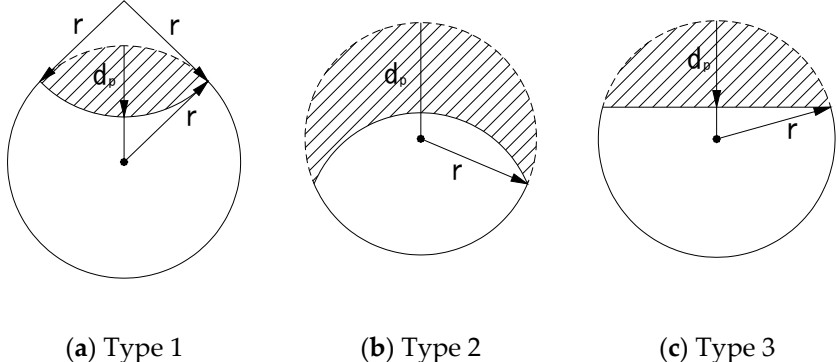

(**a**) Type 1　　　　　　　(**b**) Type 2　　　　　　　(**c**) Type 3

**Figure 4.** Three idealized pit configurations of corroded wires: (**a**) Type 1: hemispherical pit, (**b**) Type 2: concave pit, and (**c**) Type 3: planar pit.

| #1 | | | | Measured pitting depth (mm) | | | | | | | | | | | | | | | |
|---|---|---|---|---|---|---|---|---|---|---|---|---|---|---|---|---|---|---|---|
| Wire | | | Distance (mm) | 20 | 40 | 60 | 80 | 100 | 120 | 140 | 160 | 180 | 200 | 220 | 240 | 260 | 280 | 300 | |
| | 1 | Pit depth | | Invisible core wire | | | | | | | | | | | | | | | |
| | | Type | | | | | | | | | | | | | | | | | |
| | 2 | Pit depth | | | 0.18 | | | | 0.60 | | 0.46 | | | 0.17 | 0.35 | | | | |
| | | Type | | | 2 | | | | 3 | | 3 | | | 2 | 2 | | | | |
| | 3 | Pit depth | | | | | | | | 0.24 | | | | 0.23 | | 0.35 | | | |
| | | Type | | | | | | | | 2 | | | | 2 | | 2 | | | |
| | 4 | Pit depth | | | | 0.12 | | | | | 0.67 | | | | | | | 0.13 | | |
| | | Type | | | | 2 | | | | | 2 | | | | | | | 2 | | |
| | 5 | Pit depth | | | | | | | 0.54 | | | | | 0.56 | | 0.46 | | | |
| | | Type | | | | | | | 2 | | | | | 2 | | 2 | | | |
| | 6 | Pit depth | | | | 0.20 | | | 0.10 | | 0.15 | | | | 0.53 | | | | |
| | | Type | | | | 1 | | | 2 | | 2 | | | | 2 | | | | |
| | 7 | Pit depth | | | | | 0.23 | | 0.66 | 0.17 | 0.25 | | | 0.05 | 0.56 | | | | |
| | | Type | | | | | 2 | | 2 | 2 | 2 | | | 3 | 2 | | | | |

Maximum corroded section

**Figure 5.** Measured pitting depths of a corroded strand.

*2.3. Corrosion Phenomena in the External Tendon*

Figure 6 shows the profile of the voids in ducts of external tendons and the number of exposed and corroded strands counted by visual inspection for Bridges A and B. Most of the corroded strands were found in the voids, while strands in the grout in sound condition did not show any corrosion. This means that voids in ducts have a strong relationship with corrosion. The voids not filled by grout were mostly located in upper parts near air vents or anchored at high locations. A possible reason for this is that poor quality of grout causing bleeding water. It is known that bleeding water can occur due to a high w/c ratio and wicking effect. This creates voids after the bleeding water has evaporated. Another possible reason is improper grout injection process. At the time the bridges were built, there would have been no perception that air could be trapped inside due to poor regulation of grout injection. Therefore, the air vent for deflating the air was not properly positioned and was not used properly.

Grout samples from the corroded strands in Bridges A and B were analyzed for chemical content (chloride and sulfate), and the section loss due to the corrosion was measured using Equations (1)–(3) and Figure 5. Strands in areas with no grout were not considered. Figure 7a shows the relationship between the section loss and the chloride content percentage with respect to mass of concrete measured by KS F 2713 from Korean Industrial Standards (KS) [17], and Figure 7b shows the relationship between the section loss and the sulfate content measured by KS L 5120 [18]. This demonstrates that the higher the chloride content, the more severe the corrosion, while the sulfate content had a relatively smaller effect on these bridges.

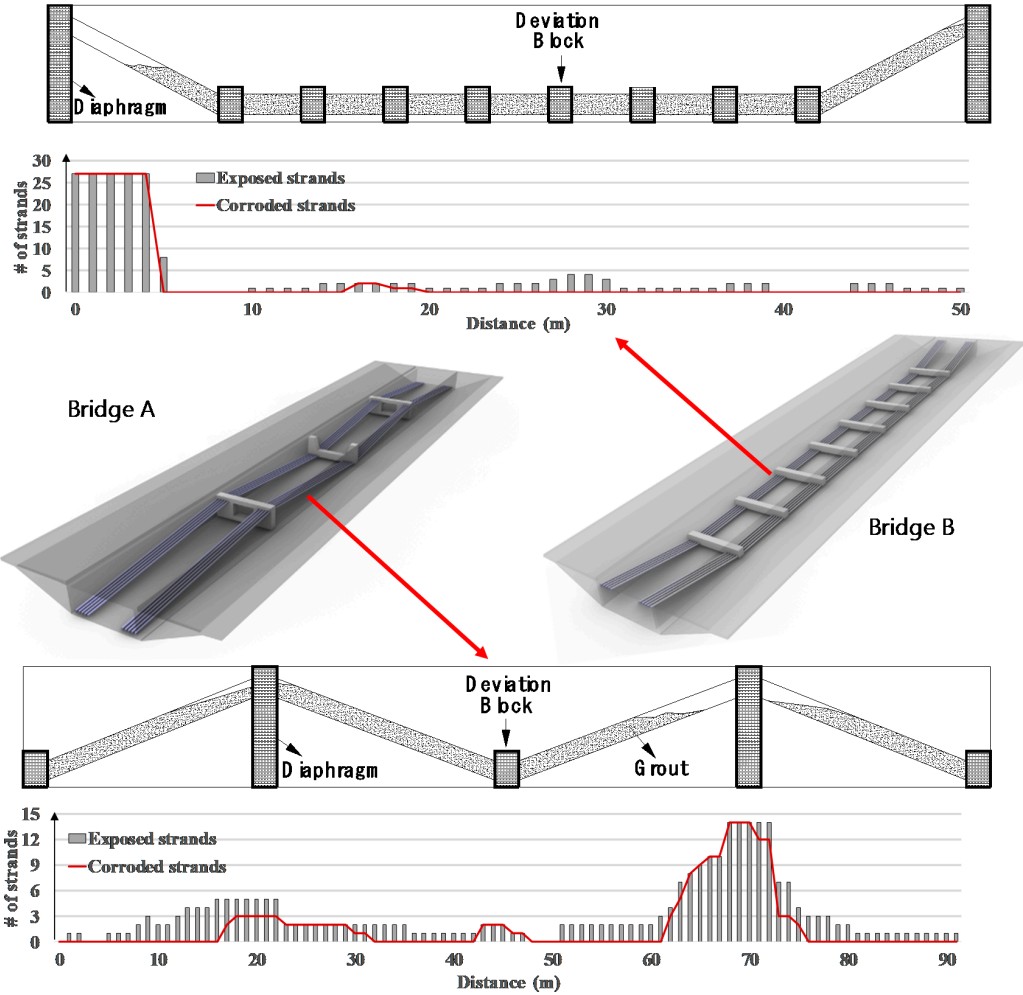

**Figure 6.** 3D model of tendon profile, state of grout, and corrosion of Bridges A and B.

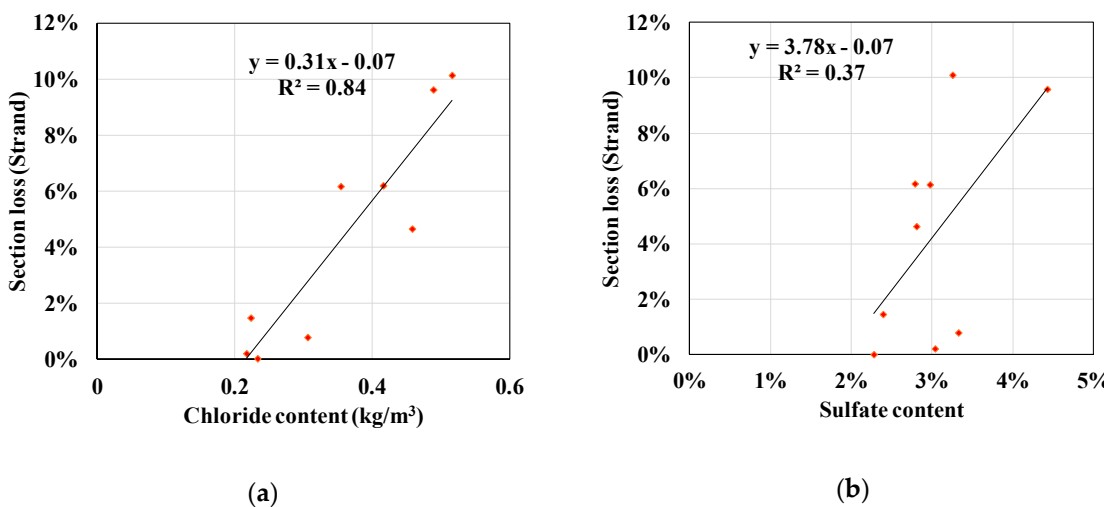

(**a**)                         (**b**)

**Figure 7.** Relationship between chemical state of grout and section loss of strands due to corrosion: (**a**) chloride content to section loss, (**b**) sulfate content to section loss.

In short, voids inside the ducts and chloride content of grouts are major factor influencing corrosion. Therefore, appropriate duct connections and proper grouting sufficient to rule out voids and protect the strands against chlorine ions would be an appropriate way to prevent corrosion.

## 3. Tensile Test of Corroded Strands

### 3.1. Test Specimens

A total of 100 corroded strands taken from Bridges A and B were investigated for corrosion. Among the specimens, a total of 86 seven-wire strands (47 strands from Bridge A and 39 strands from Bridge B) were selected for this study after exclusion of specimens that were too corroded to be used in the tensile test. Figure 8 shows two section profiles of the seven-wire strands. The strands from Bridge A had a nominal diameter of 15.2 mm and a sectional area of 138.7 mm. The diameter of the core wire was 5.2 mm, and the diameter of the outer wires was 5.0 mm. The strands from Bridge B had a nominal diameter of 12.7 mm and a sectional area of 98.7 mm. The radius of the core wire was 4.4 mm, and the radius of the outer wires was 4.15 mm. In this study, these two strands are referred to as the 15.2 type and 12.7 type, respectively.

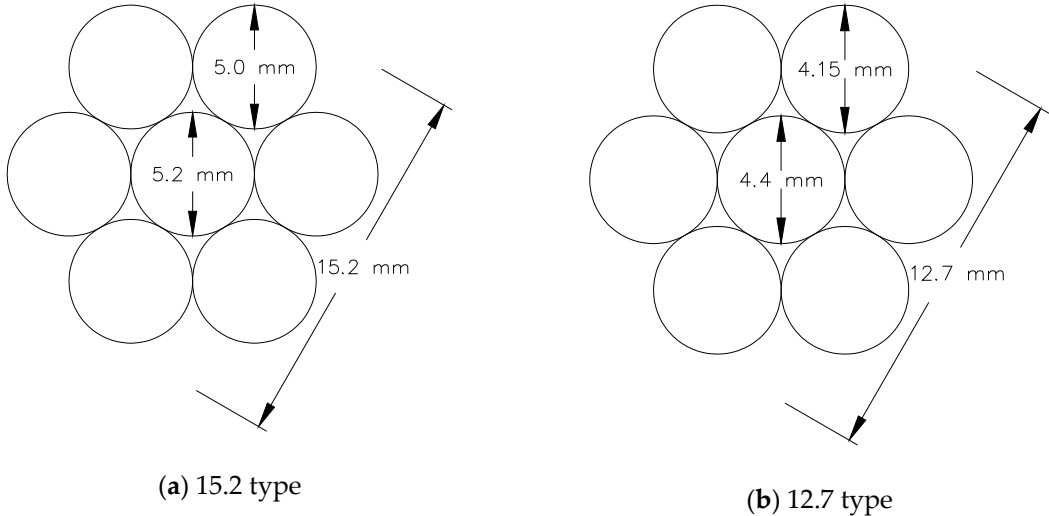

(**a**) 15.2 type　　　　　　　　　　　　　　　　(**b**) 12.7 type

**Figure 8.** Section profiles of seven-wire strands.

Before tests were conducted, the strands were cleaned with a steel brush and rust remover. An inspection to evaluate the section loss was also conducted along the length of the specimens, using a pit-depth gauge to determine the location of the most corroded section. For the quantitative corrosion comparison, the extent of section loss, denoted as η, was expressed as follows.

$$\eta = \frac{A_{sl}}{A_0} \tag{4}$$

where $A_0$ is the sectional area of the wire, and $A_{sl}$ is the loss of the sectional area from Equations (1)–(3).

After inspection of the corroded strands, it was found that the wire-unit section losses were different between strands, even though their strand-unit section losses were very similar. Figure 9 shows the relationship between wire-unit section loss of the most corroded wire and strand-unit section loss, considering the number of corroded wires. The figure also shows an example of how the data was collected. If the number of corroded wires was three, for example, the strand consisted of three corroded wires and four non-corroded wires. The maximum number of corroded wires was six because the remaining wire (core wire) was impossible to inspect.

As Figure 9 shows, it was found that strands with a higher number of corroded wires tended to have higher strand-unit section losses as well as higher wire-unit section losses. A higher strand-unit section loss is a natural result, because the strand-unit section loss is the sum of the wire-unit section losses. However, the higher levels of corrosion in the wires have a special meaning. When all wires are exposed to corrosive environmental conditions, they will corrode simultaneously, but the ducts may

have different conditions due to insufficient grout or local contaminant penetration. This can cause corrosion reactions on some wires, as shown in the section picture of Figure 9, and then other wires begin to corrode when a certain level of corrosion occurs.

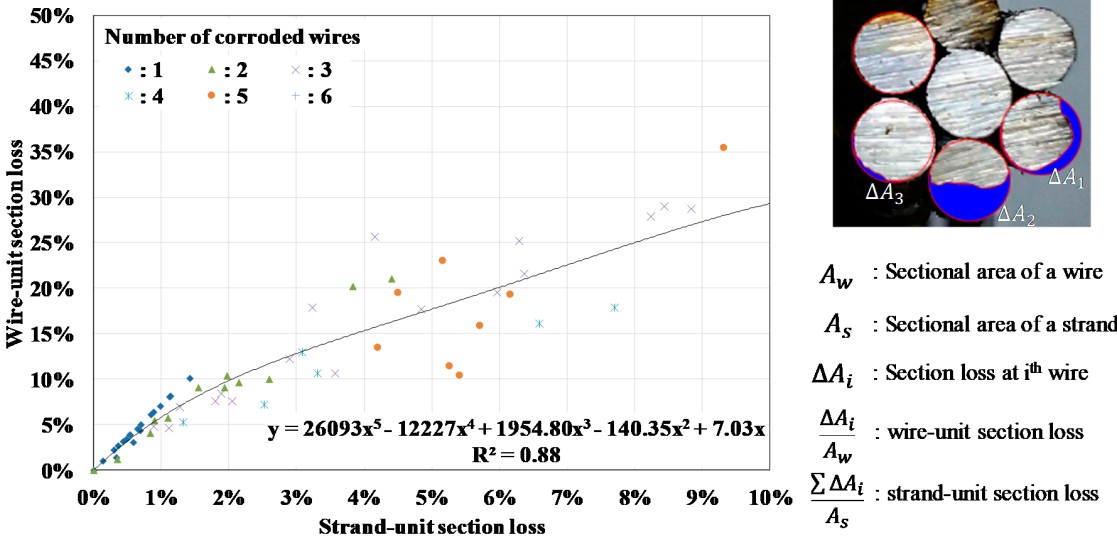

**Figure 9.** Relation between wire-unit section loss of the most corroded wire and strand-unit section loss, considering the number of corroded wires.

In addition, a higher number of corroded wires tends to produce more widely scattered data. This means that pitting corrosion can develop differently in the wires; therefore, the maximum wire-unit corrosion in a strand can be very different between corroded strands, even though they have the same level of strand-unit corrosion. If the strand fracture is defined as the moment of a single wire fracture [15], it is necessary to consider the cross-sectional loss in the wire after at least a 2% section loss in a strand.

### 3.2. Tensile Test of Corroded Strands

A tensile test was conducted as shown in Figure 10a. The specimens were loaded to failure using the displacement control method with a speed of 5 mm/min. The two ends of the strand were held by the grip. Among the strands extracted from Bridges A and B, three non-corroded strands were defined as reference specimens for the 15.2 and 12.7 types. Table 1 lists the mechanical properties of the non-corroded strands as averages of the test results from the three specimens. Where, $f_{py}$ is 0.2% proof stress of non-corroded strands, $\varepsilon_{py}$ is strain corresponding to the 0.2% proof stress, $f_{pu}$ and $\varepsilon_{pu}$ are ultimate strength and strain of non-corroded strands, respectively.

Strands with no or little corrosion showed failures of all the wires fracturing at the same moment, as shown in Figure 10b. For the corroded strands, shown in Figure 10c), the most corroded wire fractured earlier than the other wires. Owing to the fracture of the wire, the strand's strength suddenly decreased; however, the remaining wires maintained their structural capacity until the next wire was fractured, as shown in Figure 11a. Furthermore, the amount of load reduction between the moments of the first and second fractures was $\frac{1}{7}f_{pu}$, where $f_{pu}$ is ultimate strength of a corroded strand. This means that the load was evenly carried by the seven wires, and the most corroded wire fractured earlier than the others. In this study, therefore, the failure event of the strand was defined as the moment at which even one wire was broken, and the corresponding strength and strain were used to define the ultimate strength and strain.

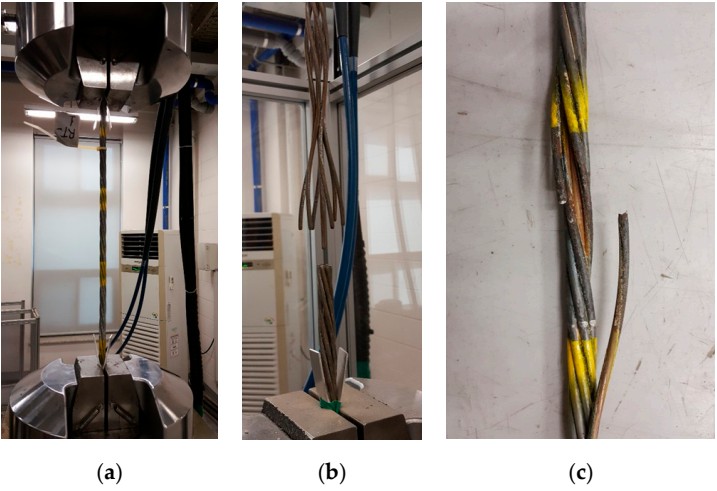

|   |   |   |
|:---:|:---:|:---:|
| (**a**) | (**b**) | (**c**) |

**Figure 10.** (**a**) Test setup, (**b**) failure of non-corroded strand, and (**c**) failure of corroded strand.

**Table 1.** Mechanical properties of the non-corroded strands.

| Strand | Nominal Sectional Area (mm²) | 0.2% Proof Stress, $f_{py}$ (MPa) | Strain Corresponding to 0.2% Proof Stress, $\varepsilon_{py}$ | Ultimate Strength, $f_{pu}$ (MPa) | Ultimate Strain, $\varepsilon_{pu}$ |
|---|---|---|---|---|---|
| 15.2 type | 138.7 | 1726 | 0.0099 | 1865 | 0.075 |
| 12.7 type | 98.71 | 1135 | 0.0090 | 1867 | 0.083 |

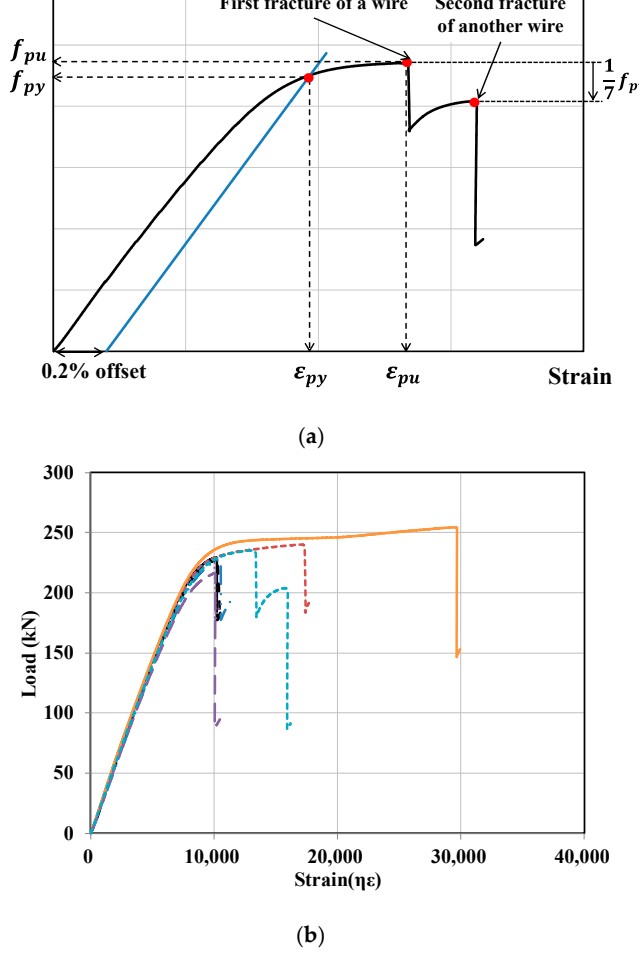

(**a**)

(**b**)

**Figure 11.** (**a**) Load–strain curve of a corroded strand and (**b**) samples from specimens.

Figure 11b shows samples of the test results from the test specimens. The elongation and maximum load due to the fracture showed significant differences among the specimens; however, the yield performances were similar. This is because even if one wire yields first due to a cross-sectional loss, the remaining wires carry the load until they yield.

### 3.3. Residual Mechanical Properties from Test Result

In this study, the residual mechanical properties due to corrosion were defined as the relation between the section loss and yield (or ultimate) strength. The section loss is explained in terms of the strand-unit and wire-unit section losses. The reason the wire-unit section loss was considered is because of the assumption that the corrosion of a strand can be localized in a few of the seven wires. A premature wire fracture due to corrosion would be the determining factor in the performance of the corresponding strand. This was validated by the decrease in strength owing to wire-unit fracture shown in the experiment (Figure 10).

ASTM A416 [19] defines the yield load as a 1% extension or 10,000 $\mu\varepsilon$, and EC2 [20] defines it as a 0.1% proof stress. However, as the extracted strand used in this research was manufactured under the KS D 7002 [21] standard, the definition of yield was a 0.2% proof stress, and the yield strength of a corroded strand was defined as a 0.2% proof stress. Specimens with severe corrosion that did not show a ductility up to a 0.2% strain rate were excluded from the yield behavior graph.

Figure 12a,b shows the relation of a 15.2 type strand's section loss with its yield strength and yield strain, respectively. Figure 12c,d shows the relation of a 12.7 type strand's section loss with its yield strength and yield strain, respectively. The coefficient of determination, $R^2$, had a maximum value of 0.0409, which indicated that the regression analysis was ineffective and the correlation of two factors was very low. This means that with the section loss of 22.52% (the most corroded specimen), the yield strength and strain showed no significant gap even under the increase in section loss, as explained in Figure 10b. However, Figure 12b, which includes numerous highly corroded strands, indicates that the yield strength was likely to decrease if the number of corroded wires and corrosion level are high. Figure 13 shows the corroded strands' yield strength and strain according to the section loss of the most corroded wire within each strand. As shown in Figure 12, the yield properties and level of corrosion from the most corroded wire also showed no significant correlation.

Before evaluating the ultimate behavior of the corroded strands, as the two strands had different ultimate strengths, to compare them, the ultimate strength was normalized through the calculation of $P_u/P_0$. $P_u$ refers to each corroded strand specimen's ultimate strength, and $P_0$ refers to the ultimate strength of a non-corroded strand. $P_0$, the ultimate loads from $f_{pu}$ in Table 1, were 261.20 kN and 184.28 kN for the 15.2 type and 12.7 type, respectively. If $P_u/P_0$ is equal to 1.0, it means the strand's strength did not decrease.

Figure 14a,b shows the relations between the strand-unit section loss and the ultimate strength and strain. As the graph indicates, unlike for the yield strength, the two factors seemed to be correlated. Furthermore, the decrease in ultimate strain could be a more serious problem than the decrease in strength. For example, for a 5% section loss, a 20% maximum strength loss can occur; however, in the case of ultimate strain, for a 5% section loss, an 85% maximum decrease in deformation capacity can occur.

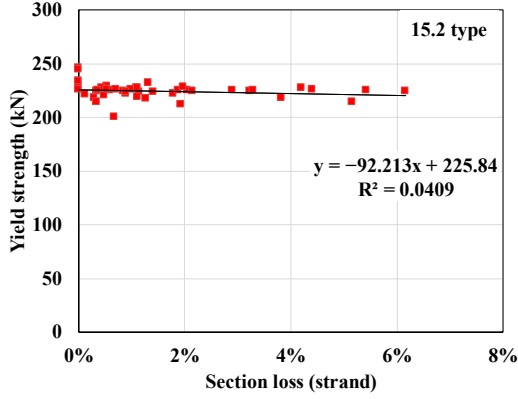

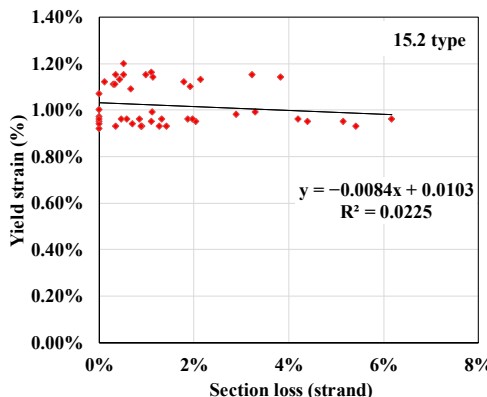

(**a**) Yield strength—Section loss relation (based on the strand)

(**b**) Yield strain—Section loss relation (based on the strand)

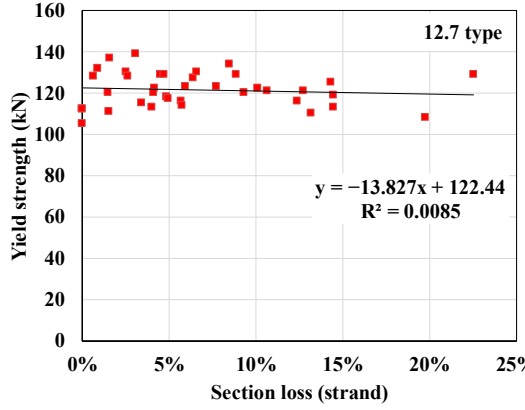

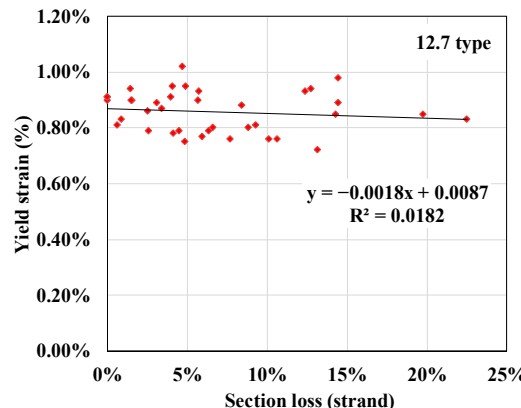

(**c**) Yield strength–section loss relationship (based on the strand).

(**d**) Yield strain–section loss relationship (based on the strand).

**Figure 12.** Yield properties of corroded strands, evaluated via section loss of strands.

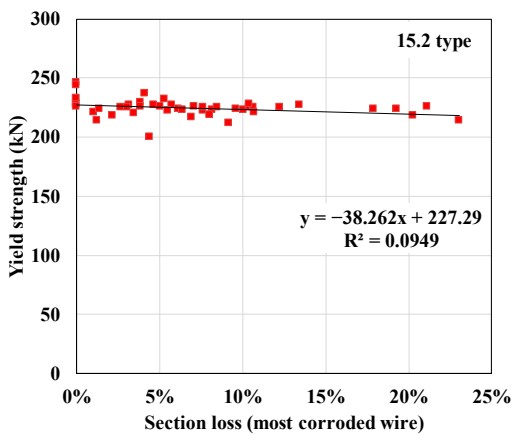

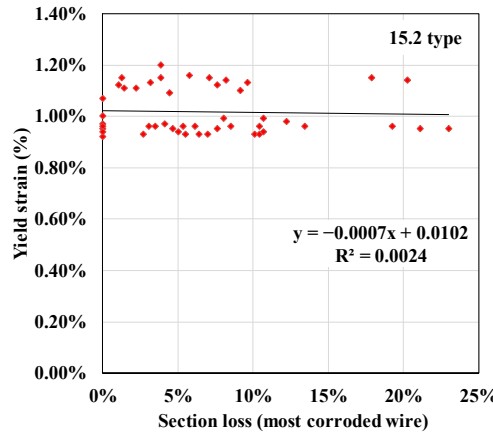

(**a**) Yield strength-section loss relation (based on the most corroded wire).

(**b**) Yield strain-section loss relation (based on the most corroded wire).

**Figure 13.** *Cont.*

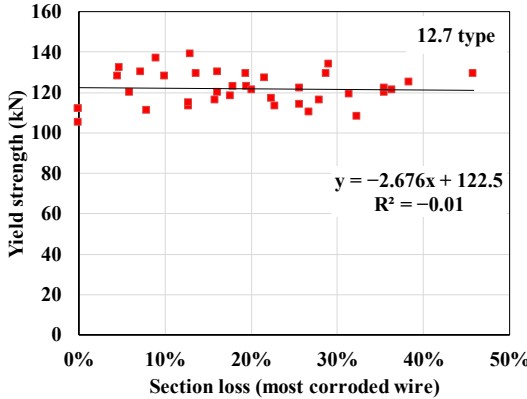

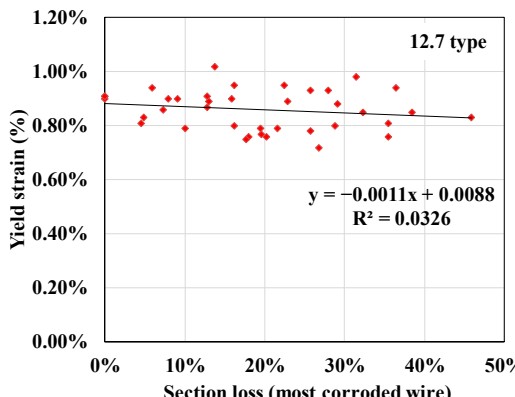

(**c**) Yield strength-section loss relationship (based on the most corroded wire).

(**d**) Yield strain-section loss relationship (based on the most corroded wire).

**Figure 13.** Yield properties of corroded strands, evaluated via the section loss of the most corroded wire in each strand.

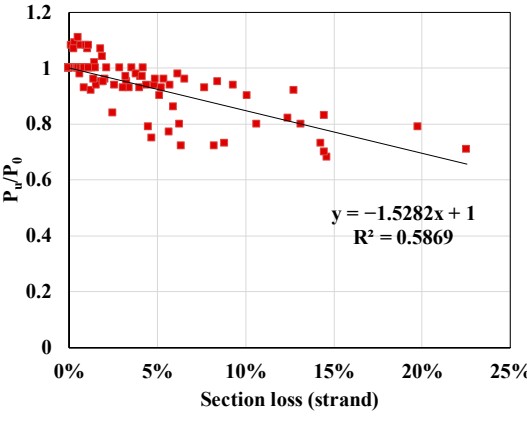

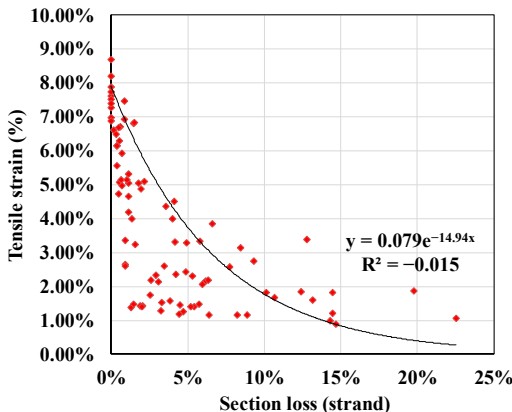

(**a**) Ultimate load–section loss relation (based on strand).

(**b**) Ultimate strain–section loss relation (based on strand).

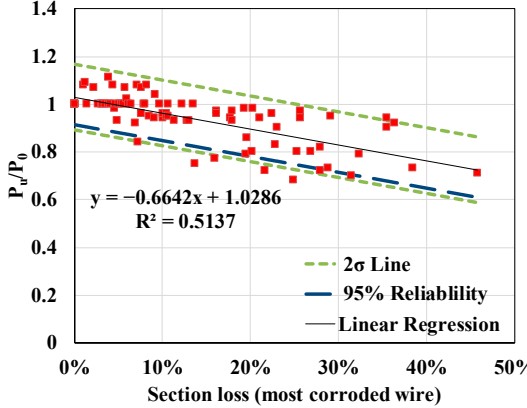

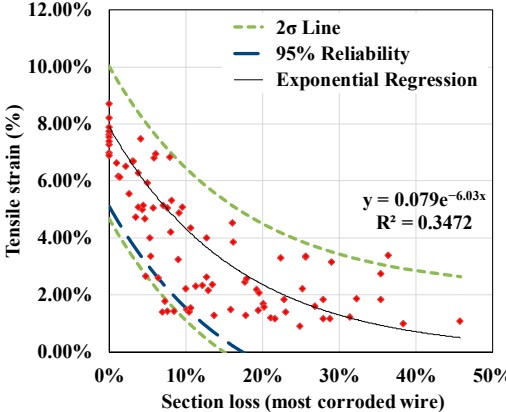

(**c**) Ultimate load–section loss relation (based on most corroded wire).

(**d**) Ultimate strain–section loss relation (based on most corroded wire).

**Figure 14.** Ultimate properties of corroded strands.

Regarding the test results, the strength of the non-corroded strands ($P_u/P_0 = 1$) and ultimate strain (7.89%, average of the ultimate strain of the two types listed in Table 1) were set as the axis and intercept, and a regression analysis was conducted. The ultimate strain showed a high deviation, implying a low credibility of the index regression analysis result ($R^2 = -0.015$). Furthermore, there were specimens that showed similar strengths and ultimate strains despite having highly varying degrees of corrosion. Such deviations in the test result distribution may have been due to highly localized corrosion pits.

Errors occurred in the process of measuring the section loss due to corrosion, and specifically in deciding the location of the maximum depth pit by inspection and in measuring due to the intercept of the near wire and curvature of the spiral wire. The stress was concentrated in the damaged part due to the irregular surface condition of the corrosion pits [22]. This is the factor that caused the most difficulty in quantifying the corrosion property. The most corroded wire can determine the behavior of the corresponding strand. This was indicated by the terraced behavior shown in Figure 10a, and it seemed more valid to measure the wire-unit section loss rather than the strand-unit section loss.

Figure 14c,d shows the section losses converted from the strand-unit to the wire-unit with the largest section loss within the strand. The explanatory power of the regression analysis increased from $R^2 = -0.015$ to $R^2 = 0.3472$. Figure 13c,d shows the lower limit curves with a 95% reliability and $\pm 2\sigma$ of the performance distribution, depicting the relations of the wire-unit section losses with the ultimate strength and strain. It was assumed that the test result followed a normal distribution with the regression analysis curve as the average.

According to lower limit curve of 95% reliability, the ultimate strength proportionally decreased in relation to the maximum corroded wire section loss. The ultimate strength values decreased by 20% and 38% when the section losses were 20% and 40%, respectively. Furthermore, the ultimate strain of the corroded strand at the lower limit curve of 95% reliability could not meet the requirement of the minimum tensile strain (3.5%) from KS D 7002 [21] if the section loss exceeded approximately 4%. The ultimate strain was evaluated as 0 when the section loss was approximately 17%.

The equations for the lower limit curve with a 95% reliability, shown in Figure 14c,d, are as follows.

$$\frac{F_{c.pu}}{F_{pu}} = -0.6642\eta + 0.913, \tag{5}$$

$$\varepsilon_{c.pu} = \varepsilon_{pu}e^{-6.03\eta} - 0.0278 \tag{6}$$

where $F_{pu}$ and $F_{c.pu}$ are the ultimate loads of the non-corroded and corroded strands in kN, respectively. $\varepsilon_{pu}$ and $\varepsilon_{c.pu}$ are the ultimate strains of the non-corroded and corroded strands, respectively. $\eta$ is the section loss according to Equation (4).

## 4. Prestressed Concrete Beams with Corroded Strands

Pape et al. (2013) [23] conducted flexural test on 45 years old corroded prestressed concrete beams. The test result showed significant loss of ultimate capacity associated with degree of corrosion, and current design theory with actual material properties, free of cracking, and corrosion damage could overestimate the actual capacity. Therefore, a method of ultimate strength evaluation on PSC beams is suggested herein, using the reduced material properties of corroded strands mentioned above.

### 4.1. Test Specimen

To observe the behavior of the corroded strands in a prestressed concrete member, four simply supported beam specimens were manufactured and tested. The dimensions and details of the specimens are presented in Figure 15. The post tensioned beams had a section of $150 \times 220 \times 2000$ mm with a single seven-wire $\Phi15.2$ mm strand.

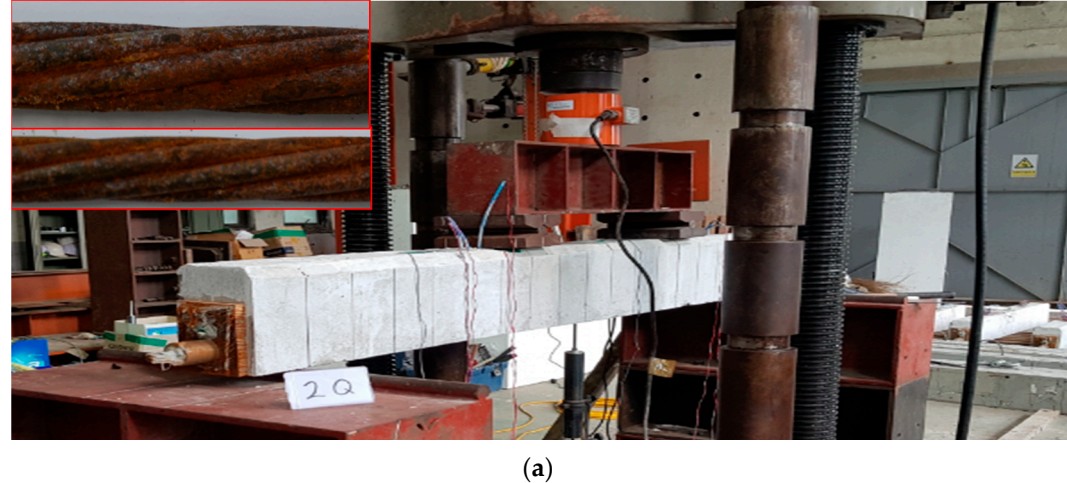

(**a**)

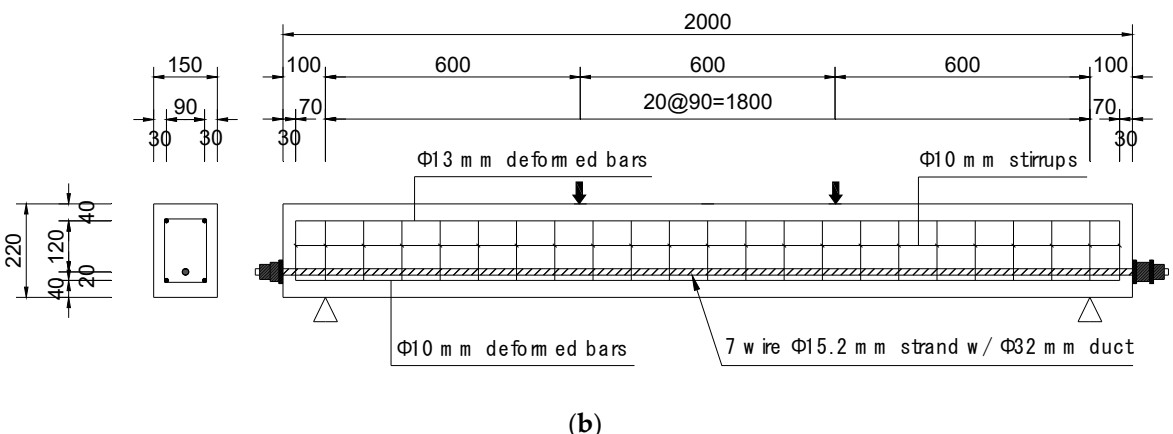

(**b**)

**Figure 15.** (**a**) Test setup and (**b**) detail drawing.

Table 2 lists the information on the material and strands' corrosion status for the four manufactured PSC beam specimens. Strands used in the test beams were artificially corroded through exposure to NaCl (to create humid and dry conditions in cycles) for several months, as shown in Figure 15a. The corroded strands had very similar corrosion shape to the actual corrosion shapes observed from existing Bridge B (Figure 2) in the 0–10% range of section loss. Before the corroded strands were introduced to the test beams, section loss was evaluated by Equations (1)–(3). The RB specimen ID refers to the reference member without corrosion. QB refers to a member with corrosion at 1/4 of the span length (450 mm from the support), and MB refers to a member with corrosion in the center of the span length.

**Table 2.** Beam test specimens.

| ID | Location of Corrosion | Section Loss (Strand) | Section Loss (Most Corroded Wire) | Effective Prestressing Force | Material Properties |
|---|---|---|---|---|---|
| **RB** | - | 0% | 0% | | Concrete: 44.07 MPa |
| **QB** | 1/4 | 4.82% | 21.56%. | 160 kN | Yield strength (rebar): 400 MPa |
| **MB1** | 1/2 | 6.67% | 22.24% | | Tensile strength (rebar): 560 MPa |
| **MB2** | 1/2 | 7.51% | 18.89% | | Yield strength (strand): 1630 MPa |
| | | | | | Tensile strength (strand): 1865 MPa |
| | | | | | Elastic modulus of strand: 194.5 GPa |

The average yield, ultimate strength, and elastic modulus of the prestressing steel from the tensile test result were 1630 MPa, 1865 MPa, and 194.5 GPa, respectively. The characteristic yield and ultimate strength of the deformed bars were 400 MPa and 560 MPa, respectively. The concrete was manufactured to have a compressive strength of 40 MPa, and the average compressive strength was 44.07 MPa from concrete cylinder tests when the experiment was conducted. Regardless of corrosion status, the effective prestress force on the strands was equally applied as 160 kN.

Flexural tests were performed using the displacement control method under a four point bending condition. The span length of the specimen was 1800 mm, and the loading position was 600 mm from a support. Loading was added until the compressed concrete reached a compressive strain of 0.003, even if the strand fractured earlier due to corrosion.

### 4.2. Test Result

Figure 16 shows the load–displacement curves of the specimens and an image of the wire fractures after the tests were conducted. Table 3 summarizes the strength, displacement, and failure modes of the specimens. Considering the material properties in Table 2, the non-corroded RB's nominal strength was 159.3 kN. The strength and displacement in the experiment were 163.9 kN and 18.40 mm, respectively. After the yielding of the strand and rebar, the compressed concrete reached a crushing failure mode. After the QB member experienced one wire fracture, it then reached a crushing failure. The strength and displacement at the moment of fracture were 161.0 kN and 17.89 mm, respectively. The corrosion reduced the strength and flexural capacity by 1.77% and 2.77% compared to the RB member. This means that the corroded strand had a limited influence on the strength of the member due to sufficient bond between the strand and surrounding grout.

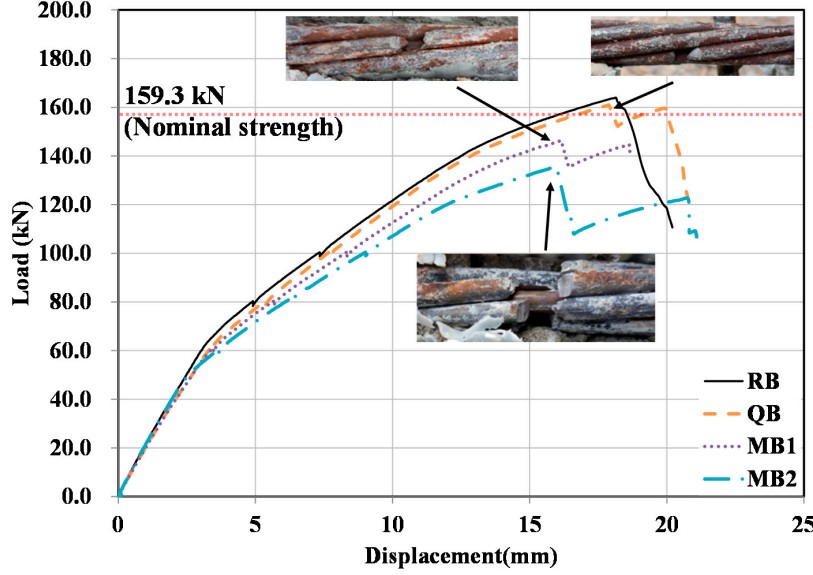

**Figure 16.** Load–displacement curves of PSC beam specimens.

**Table 3.** Test results of PSC beam specimens.

| ID | Maximum Load (kN) | Load Reduction Ratio to RB | Maximum Disp. (mm) | Disp. Reduction Ratio to RB | Failure Mode |
|---|---|---|---|---|---|
| RB | 163.9 | - | 18.40 | - | Compressive concrete crush |
| QB | 161.0 | 1.77% | 17.89 | 2.77% | Compressive concrete crush after one wire fracture |
| MB1 | 146.4 | 10.68% | 16.14 | 12.28% | Compressive concrete crush after two wire fractures |
| MB2 | 135.6 | 17.27% | 15.98 | 13.15% | Compressive concrete crush after two wire fractures |

MB1 and MB2, which had corrosion in the center of the span length, showed strength decreases of 10.68% and 17.27% and decreases in the displacement capacity of 12.28% and 13.15%, respectively. Both specimens experienced a compression failure after two wire fractures. Additionally, four wires failed in MB2, which means that several wires can fail at one time. The corrosion in the strands at the location of the maximum moment significantly influences the ultimate behavior of a PSC beam.

### 4.3. Determination of the Flexural Strength of a Corroded PSC Beam

To calculate the flexural strength of a PSC beam, the approximate value of $f_{ps}$, which refers to the stress of the strand at the moment of the beam's failure, can be obtained from the design codes, or the strain compatibility method can be utilized to derive the value. According to ACI 318-14 [24], Equation (7) is the approximate expression for $f_{ps}$. $f_{c.ps}$ and $\rho_{c.p}$ refer to the ultimate strength of the corroded strand and steel ratio, respectively. The section loss by corrosion can be calculated from Equations (1)–(3), and the decreased ultimate strength can be calculated using Equations (4) and (5), along with the result of tensile tests given in Figure 17.

$$f_{c.ps} = f_{c.pu} \left[ 1 - \frac{\gamma_p}{\beta_1} \left\{ \rho_{c.p} \frac{f_{c.pu}}{f_{ck}} + \frac{d}{d_p} (\omega - \omega') \right\} \right],$$

(7)

where

$f_{c.ps}$: stress of the corroded prestressing strand when the beam fails;

$f_{c.pu}$: ultimate stress of the corroded prestressing strand;

$f_{c.pu}$: coefficient according to the type of strand $\begin{cases} 0.55 \text{ when } f_{py}/f_{pu} \geq 0.80 \\ 0.40 \text{ when } f_{py}/f_{pu} \geq 0.85 \\ 0.28 \text{ when } f_{py}/f_{pu} \geq 0.90 \end{cases}$ ;

$\beta_1$: ratio between the depth of an equivalent rectangular concrete stress block and the neutral axis depth (= a/c);

$\rho_{c.p}$: corroded prestressing steel ratio $\left( = A_{c.p}/bd_p \right)$;

$d$: depth of tensile reinforcement;

$d_p$: depth of prestressing strand;

$\omega$: tensile reinforcement index $\left( = \rho \frac{f_y}{f_{ck}}, \ \rho = \frac{A_s}{bd} \right)$;

$\omega'$: compression reinforcement index $\left( = \rho' \frac{f_y}{f_{ck}}, \ \rho' = \frac{A_s'}{bd} \right)$.

$f_{c.ps}$ from Equation (7) for each specimen and the corresponding flexural strength are listed in Table 4. The RB, MB1, and MB2 specimens showed reasonable estimations of the strength, while QB showed a value different from the test value. The reason is that the cross-section of corrosion was at 1/4 of the span length, such that a lesser flexural moment was carried than that occurring at the center. Therefore, the governing cross-section for the nominal flexural strength became the center (154.47 kN) rather than 1/4 (211.73 kN), even though the reduced $f_{c.ps}$ is considered. The failure mode, however, showed a wire fracture at the 1/4 of the span length. This means that the reduced ultimate strain of the corroded strand could be the governing factor of the flexural strength. To consider this, the strain compatibility method should be used.

Although MB2 showed a lower flexural strength of 135.6 kN than that of MB1 at 146.4 kN, the flexural strength calculated by Equation (7) was higher (138.53 kN < 140.6 kN). This was because MB2's maximum wire-unit section loss of the corroded strand was lower than that of MB1 (18.89% < 22.24%), making the $f_{ps}$ from Equation (7) a lower value than that of MB1 (1289 MPa < 1327 MPa). This phenomenon was caused by an uncertainty in the tensile behavior as well as an undefined bonding behavior of the corroded strand.

For issues regarding the residual service life of an entire PSC bridge with respect to corrosion, the flexural strength prediction method mentioned above is not enough. Section loss and stress concentration due to corrosion results in change of stress conditions in a bridge. The residual service life is estimated by serviceability limit state in terms of stress limits at the joints of post-tensioned

box girders. Ultimate limit state needs to be checked considering strength reduction and stress concentration. Therefore, further study is required for the residual service life.

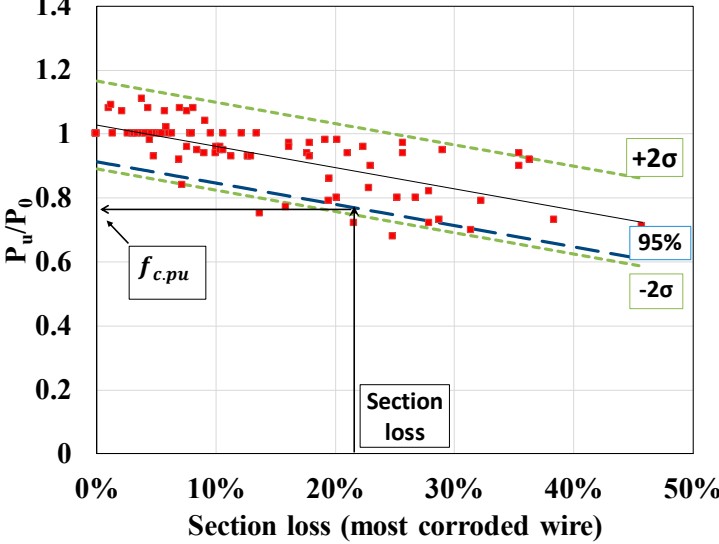

**Figure 17.** Determination of $f_{c.ps}$ from 95% lower reliability limit of ultimate strength.

**Table 4.** Flexural strength evaluation using approximated determination of $f_{ps}$ with Equation (7).

| ID | Ultimate Strength of Corroded Strand Based on 95% Lower Reliability Limit | $f_{c.ps}$ from Equation (7) | Maximum Load (Test) | Maximum Load Considering Equation (7) | Difference |
|---|---|---|---|---|---|
| RB | 1860 MPa | 1614 MPa | 163.9 kN | 154.47 kN | −5.75% |
| QB | 1432 MPa | 1293 MPa | 161.0 kN | 154.47 (211.73 *) kN | −4.06% |
| MB1 | 1423 MPa | 1289 MPa | 146.4 kN | 138.53 kN | −5.38% |
| MB2 | 1469 MPa | 1327 MPa | 135.6 kN | 140.6 kN | 3.69% |

* Ultimate load for the section at point of corrosion (1/4 of span length).

## 5. Conclusions

In this study, a total of 86 seven-wire strands taken from existing post-tensioned segmental box girder bridges were investigated for section loss due to corrosion. The mechanical properties of the corroded strands derived from a 95% lower reliability curve were estimated by tensile tests. The flexural behaviors of four corroded PSC beams with different levels and locations of corrosion were also introduced. A method to evaluate flexural strength using an approximated $f_{c.ps}$ calculation was suggested. The conclusions of this study are as follows.

(1) A higher number of corroded wires in one strand tends to be associated with a higher level of corrosion (section loss) in the wire. This indicates that the corrosion reactions continue to occur in some wires, and once a certain level of corrosion occurs, other wires also begin to corrode. Because a plurality of strands is bundled together in one tendon due to tension, the inner wires are protected from corrosive conditions as compared to the outermost wires; therefore, inner wires corrode sequentially after the outer wires are sufficiently corroded.

(2) The maximum wire-unit corrosion in a strand can be very different between corroded strands, even though they have the same level of strand-unit corrosion. If the strand fracture is defined as the moment at which a single wire fractures, it is necessary to consider the cross-sectional loss in the wire, after at least a 2% section loss of a strand.

(3)    Within the wire-unit section loss of 45.79% and the strand-unit section loss of 22.52% (the most corroded specimen among the 86 specimens), the yield strength and strain showed no significant gap even under the increase of section loss, because even if one wire reaches a yield first due to cross-sectional loss, the remaining wires can carry the load until they yield. However, the yield strength is likely to decrease if the number of corroded wires and the corrosion level are high.

(4)    The ultimate strength and strain for the corroded strand were introduced by the lower limit curve of 95% reliability from the tensile test result. The ultimate strength proportionally decreased; the ultimate strength was reduced by 39.11% if the wire-unit section loss was 45.79% based on the lower limit curve of 95% reliability. The ultimate strain of the corroded strand could not meet the requirement of minimum tensile strain (3.5%) from KS D 7002 [21] if the wire-unit section loss exceeded approximately 4%. Formulas for the lower limit curve of 95% reliability were also introduced.

(5)    Based on the test results of the PSC beam specimens with a single corroded strand, a method to evaluate the flexural strength using $f_{c.ps}$ was suggested. The proposed method showed an approximate estimate of flexural strength with a difference of 4.72% on average; however, it is limited to considering the rapid reduction of the ultimate strain of corroded strands.

A further study is required for the following issues. This study considered the single-strand PSC beam. To use mechanical properties of corroded strands for flexural strength of PSC beams, it was assumed that effective prestress had no additional loss and no stress redistribution between the strands. However, this assumption may be inaccurate to flexural behavior of multi-strand PSC beams. In addition, flexural performance of corroded PSC beams can be defined by not only strength but also ductility. Tensile test of corroded strands showed that the strain decreased rapidly with corrosion, so corroded PSC beams may have ductility problems. A strain-compatibility method could be adopted to evaluate both performances.

**Author Contributions:** Conceptualization, C.-S.S.; methodology, C.-S.S. and C.-H.J.; validation, C.-H.J.; formal analysis, C.-H.J.; investigation, C.-H.J. and C.D.N.; resources, C.-H.J. and C.D.N.; data curation, C.-H.J.; writing—original draft preparation, C.-H.J.; writing—review and editing, C.-H.J.; visualization, C.-H.J.; supervision, C.-S.S; project administration, C.-H.J.; funding acquisition, C.-S.S. All authors have read and agreed to the published version of the manuscript.

**Funding:** This study was funded by the Ministry of Land, Infrastructure and Transport (MOLIT) of the Korean government and the Korea Agency for Infrastructure Technology Advancement (KAIA).

**Acknowledgments:** This study was supported by a grant (18SCIP-B128570-02) from the Smart Civil Infrastructure Research Program funded by the Ministry of Land, Infrastructure and Transport (MOLIT) of the Korean government and the Korea Agency for Infrastructure Technology Advancement (KAIA), and this study was also supported by the Chung-Ang University Young Scientist Scholarship in 2018.

**Conflicts of Interest:** The authors declare no conflict of interest.

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
