# Peer review of "Assessment of Mechanical Properties of Corroded Prestressing Strands"

_applsci, doi:10.3390/app10124055_

Round 1
Reviewer 1 Report
Article review
Applied sciences journal, MDPI
'Assessment of mechanical properties of corroded prestressing strands'
This is an interesting paper that considers the assessment of the mechanical properties of corroded prestressing strands and their influence on structural behaviour.
The paper states that corrosion induced failure is not considered in the design of prestressed concrete structures and as a result of this sudden collapse may follow. The paper also states that it is a difficult task to quantify both the amount of corrosion that has occurred as well as its location within a given structure. A method to evaluate the level of corrosion and the corresponding mechanical properties of the corroded strands is needed.
I would suggest the following amendments are made to the paper in order to clarify various terms and help the reader follow the current text and discussions therein.
Line 45: Figure 1 image (a): Identical picture has been previously published in reference [15] and should be replaced. Not referenced to [15].
Line 47: 'inspections' should read 'inspection'.
Line 47: 'mostly' should appear after 'have'.
Line 47: 'because' should be replaced with 'as'.
Line 48: after 'replace' insert 'these'
Line 48: 'it is' should be replaced with 'these are'
Line 48: 'if' should be replaced with 'when'
Line 48: 'when' should be inserted after 'tendons'
Line 49: 'introduced' should be replaced with 'mentioned'
Line 49: 'its' should be replaced with 'their'
Line 52: 'The first method....' should be starting as a new paragraph
Line 61: delete text 'methods to evaluate pitting'
Line 61: Insert 'the methods to evaluate pitting given in' after the word 'using'
Line 61: after '[8]' should read 'regarding corrosion'
Line 63: Insert 'have' after '[9-14]'
Line 77: Insert after 'bridges' 'referred to as 'A' and 'B'
Line 77: Insert after 'in' 'an'
Line 77: Insert after 'that' 'were known to'
Line 79: after 'A fracture' Bridge 'A' or Bridge 'B' should be mentioned
I would suggest that a cross-section be provided for both Bridges 'A' and 'B' in order to make this clearer.
Line 81: replace 'are' with 'were'
Line 84: Not clear what 'air bents' refers to? This should be amended and or replaced
Line 88: Delete 'the'
Line 91: Most of this line should be redrafted to read 'Owing to corrosion in the strands occurring along their length, the pitting was very irregular, making......'
Line 105: insert [15] after ' previous research'
Line 116: Figure 2: Identical picture has been previously published in the reference [15] and should be replaced with a different one.
Line 117: Figure 3: location should be identified on a section sketch
Line 118: Figure 4: near identical figure has been previously published in reference [15] and should be replaced with a different one.
Line 120: Figure 5: 'Distance' should have units. Wire numbers should be identified on a sectional sketch
Line 124: Ditto comments for line 84
Line 136: Figure 6: Sketches should identify which is Bridge 'A' and which is Bridge 'B' with (a) and (b). Lengths should also be included.
Line 137: Figure 7: gap missing between loss and (strand)
Lines 140 and 141: Delete lines after '[15]' or redraft
Lines 143 to 147: This information should be provided with a sketch
Line 151: Equation (7) shows AO' but AO defined in the text. This should be amended
Line 154: Insert word 'the' after 'of' and 'that'
Lines 155 to 158: Starting with 'The' on line 155 should be redrafted due to lack of clarity and or poor use of English
Lines 163 to 170: A sketch and or definitions should be provided or passage should be redrafted due to lack of clarity of meaning and or poor use of English.
Line 178: Figure 8: not clear what is being presented. Suggest that a sketch is provided to accompany
Line 181: delete 'their'
Line 182: delete 'and' after 5 mm/min and provide a full stop. Start new line with 'The'
Line 183: 'the bridge' but not identified as 'A' or 'B'?
Line 191: fpu not defined in the line
Line 203: Figure 9 image (a): previously published in reference [15] and should be replaced
Line 207: delete the 's' from relations so as to read relation
Line 208: insert 'the' after 'between'
Line 295: delete 'at a certain location'
Lines 303 and 304: 'bar' should be 'bars'. The yield and ultimate strengths are given as 400 MPa and 560 MPa, respectively. It is not clear how these values were obtained this should be clarified and included in the text.
Line 308: 4-point loading mentioned but Figure 14 (a) and (b) appear to show 2-point loading? This should be reviewed
Line 311: 'crushing state' mentioned but no definition of this is has been provided?
Line 313: Figure 14 (b) should be larger so that details can be seen. Not clear what '10 @ 90 stirrups' is? This should be clarified.
Lines 314 and 315, Table 2: Not clear if the material properties are characteristic values or values determined from tests? See comments for lines 303 and 304.
Line 335: Table 3: Failure modes should be accompanied by images.
Lines 394 to 397 are a repeat of lines 387 to 390
Line 402: section loss % should be quoted to the same accuracy as lines 409 etc.
Lines 418 to 421 should be redrafted as meaning is not clear.
Lines 431 to 433 are a repeat of lines 428 to 430. One of these should be modified.
The Abstract should be reviewed and amended in the light of the above. Further, a sketch or definition of what is meant by 'external tendons in existing bridges' should be provided in the context of this paper. This is confusing as ducting and grouting is mentioned in the current text and it is not clear whether pre-stressing, post-tensioned or pre-tensioning is being referred to? This should also be clarified.
The authors may wish to look at: Torill Pape and Robert Melchers paper called 'Performance of 45-year-old corroded prestressed concrete published in ICE proceedings, Structures and Buildings, vol 166, issue SB10, 2013.
Unfortunately, there are many areas of this paper that need improvement for me to recommend publication in its current form. I Trust that the authors find the above comments and suggestions useful when amending their article.
Reviewer 2 Report
The paper is interesting and can be published in the Applied Sciences with the following improvements and corrections:
- Please avoid the same text, equations (1-6), photos (for example Fig. 1, 2, etc) and drawings which were published by authors in previous papers. The authors can cite this information accordingly.
- Based on the above, the authors should clearly be emphasised what is new in relation to their previous papers.
- The photo or drawings of the bridges A and B are needed.
- Fig. 6 seems to be key and can be presented in a more readable way. Now, it is difficult to understand the authors' way of thinking.
- The authors should also add the method of protection of strands against corrosion.
- The residual service life of the strands and the entire bridges should be also presented. How does the strand corrosion affect the service life of the bridge?
Reviewer 3 Report
This is a well-conducted work.
Minor suggestions include:
- Typing equations 1-6 instead of inserting them as Figures.
- Figure 2 as a table rather than an image.
Author Response
Thank you for your review.
C1) Typing equations 1-6 instead of inserting them as Figures.
A) The equations are revised and inserted as typing equations.
C2) Figure 2 as a table rather than an image
A) I am afraid that I can not get your point. The figure 2 is showing how a pit depth gauge is used. Isn't it about Figure 3?
Round 2
Reviewer 1 Report
Figure 2 is still very similar, near identical, to the previously published image. This should be deleted completely if no other image is available and reference to [15] should be made.
Author Response
The figure 2 is replaced with a figure so that avoids similarity.
Reviewer 2 Report
The authors well responded to the reviewer comments.
Author Response
Thank you for the review.